# Ferroptosis as a Therapeutic Avenue in Triple-Negative Breast Cancer: Mechanistic Insights and Prognostic Potential

**DOI:** 10.3390/biomedicines13082037

**Published:** 2025-08-21

**Authors:** Taimoor Riaz, Muhammad Ali Saleem, Muhammad Umar Muzafar Khan, Muhammad Abdul Rehman Rashid, Muhammad Zubair

**Affiliations:** Department of Bioinformatics and Biotechnology, Government College University Faisalabad, Faisalabad 38000, Pakistan; taimoor.201830243@gcuf.edu.pk (T.R.); alisaleem3u@gmail.com (M.A.S.); umarmuzafark@gmail.com (M.U.M.K.)

**Keywords:** IFNG, prognostic biomarker, precision oncology, translational research, gene expression profiling, bioinformatics analysis

## Abstract

**Background and Aims:** Triple-negative breast cancer (TNBC) is a clinically aggressive malignancy marked by rapid disease progression, limited therapeutic avenues, and high recurrence risk. Ferroptosis an iron-dependent, lipid peroxidation-driven form of regulated cell death that has emerged as a promising therapeutic vulnerability in oncology. This study delineates the ferroptosis-associated molecular architecture of TNBC to identify key regulatory genes with prognostic and translational significance. **Methods:** Transcriptomic profiles from the GSE103091 dataset (130 TNBC and 30 normal breast tissue samples) were analyzed to identify ferroptosis-related differentially expressed genes (DEGs) using GEO2R. Protein–protein interaction (PPI) networks were constructed via STRING and GeneMANIA, with functional enrichment performed through Gene Ontology (GO), Kyoto Encyclopedia of Genes and Genomes (KEGG), and Reactome analyses. Prognostic relevance was evaluated using GEPIA, BC-GenExMiner, and Kaplan–Meier Plotter survival analyses. **Results:** Six ferroptosis drivers (MAPK1, TLR4, IFNG, ATM, ULK2, and ATF3) and five suppressors (NFS1, GCLC, TP63, CD44, and SRC) were identified alongside HMOX1, a bifunctional regulator with context-dependent pro- and anti-ferroptotic activity. Enrichment analyses revealed significant associations with oxidative stress regulation, autophagy, immune modulation, and tumor progression pathways. Elevated IFNG expression was consistently linked to improve overall, disease-free, and distant metastasis-free survival, underscoring its dual function in antitumor immunity and ferroptosis sensitization. **Conclusions:** Ferroptosis represents a critical axis in TNBC pathophysiology, with IFNG emerging as both a prognostic biomarker and a viable therapeutic target. These insights provide a mechanistic foundation for integrating ferroptosis-inducing agents with immunotherapeutic modalities to enhance clinical outcomes and overcome therapeutic resistance in TNBC.

## 1. Introduction

Triple-negative breast cancer (TNBC) is a biologically aggressive subtype of breast cancer defined by the absence of estrogen receptor (ER), progesterone receptor (PR), and human epidermal growth factor receptor 2 (HER2) expression [1]. It accounts for approximately 15% of all breast cancer cases and is frequently associated with younger patient age, higher histological grade, and early metastatic spread particularly to visceral organs and the central nervous system. The lack of hormone and HER2 receptors renders TNBC unresponsive to targeted endocrine and anti-HER2 therapies, thereby limiting treatment options to cytotoxic chemotherapy [2]. Unfortunately, chemotherapy often yields suboptimal outcomes due to intrinsic or acquired resistance, resulting in high rates of recurrence and a five-year survival rate that remains among the poorest of all breast cancer subtypes.

TNBC is also notable for its substantial molecular and clinical heterogeneity. Advances in transcriptomic profiling have revealed distinct molecular subtypes within TNBC, including basal-like, mesenchymal, luminal androgen receptor-positive, and immune-enriched subgroups. These classifications reflect unique underlying biology, immune landscapes, and treatment responses, complicating clinical management but also presenting opportunities for more personalized therapeutic approaches [3]. The immune-enriched subtype, in particular, has shown potential responsiveness to immune checkpoint inhibitors, though durable benefit is limited by the tumor’s capacity for immune evasion and microenvironmental remodeling [4].

In this context, ferroptosis, a regulated form of cell death driven by iron-dependent lipid peroxidation, has gained increasing attention as a therapeutic vulnerability in cancer. Unlike apoptosis or necrosis, ferroptosis is initiated by the accumulation of reactive oxygen species (ROS) and lipid hydroperoxides due to dysregulation in iron metabolism and antioxidant defense [5].

Key regulators of ferroptosis include glutathione peroxidase 4 (GPX4), the cystine/glutamate antiporter SLC7A11, and ferritin-associated proteins. Intriguingly, recent studies have suggested that ferroptosis can interact with immune responses in the tumor microenvironment (TME), thereby offering dual benefit as both a direct cytotoxic strategy and an immunomodulatory mechanism [6,7].

Interferon-gamma (IFNG), a type II interferon produced predominantly by activated T lymphocytes and natural killer (NK) cells, plays a central role in antitumor immunity. It enhances antigen presentation via upregulation of MHC molecules, activates cytotoxic T cells, and modulates the TME to support immune infiltration [8]. In addition to its established immunologic functions, IFNG has recently been linked to ferroptotic regulation [9]. Notably, IFNG has been shown to sensitize tumor cells to ferroptosis by downregulating SLC7A11 expression and enhancing lipid peroxidation [10]. This suggests that IFNG may serve as a molecular bridge between immune activation and ferroptotic vulnerability, particularly relevant in highly immunogenic tumors such as TNBC. However, the prognostic value of IFNG and its mechanistic contributions to ferroptosis in TNBC remain insufficiently explored.

In this study, we present an integrative bioinformatics approach to investigate the expression profile, prognostic relevance, and ferroptosis-related network of IFNG in TNBC. Utilizing the publicly available GSE103091 transcriptomic dataset, we conducted differential gene expression analysis, protein–protein interaction (PPI) network construction, pathway enrichment analysis, and survival validation across multiple clinical platforms [11]. By focusing on IFNG’s regulatory context within ferroptosis-associated signaling, our study aims to provide mechanistic insights that support the development of IFNG as both a prognostic biomarker and a candidate for immune-ferroptotic therapeutic strategies in TNBC.

## 2. Materials and Methods

### 2.1. Dataset Acquisition and Differential Expression Analysis

The GSE103091 dataset was obtained from the Gene Expression Omnibus (GEO) database, a publicly available resource that houses high-throughput sequencing and microarray data. We selected this dataset because it was specific to triple-negative breast cancer (TNBC) and contained data from 130 TNBC patients and 20 healthy controls, thus making it very suitable for discovering TNBC-specific biomarkers. DEGs were determined using GEO2R, a web-based easy-to-use analysis tool, in GEO (https://www.ncbi.nlm.nih.gov/geo/geo2r/) accessed on 17 October 2024 [12]. GEO2R allows users to compare the different day experimental conditions (controls vs. TNBC in this case) and employs the well-established statistical package known as limma (Linear Models for Microarray Data) for computing output fold change and *p*-values.

The parameters for DEG selection included the adjustment *p*-value to ensure statistical significance and biological utility. This primary objective of this analysis was to determine genes that were significantly upregulated or downregulated in TNBC relative to healthy controls. DEGs served as the basis for interaction analysis and pathway enrichment analysis.

### 2.2. Protein–Protein Interaction Network Analysis

A protein–protein interaction (PPI) network analysis was performed to enrich the functional relations among DEGs in TNBC. This analysis outlines the important proteins and pathways involved in TNBC development and potential for therapeutic agents. The STRING and GeneMANIA were used to build and assess the PPI networks. STRING integrates known and predicted protein–protein interactions derived from different sources, including experimental studies, curated databases, co-expression analyses, and computational predictions (https://string-db.org/) accessed on 17 October 2024 [13]. It also provided a confidence score for each interaction, allowing for prioritization of high-confidence associations. GeneMANIA performed this analysis by extending the network of characterized proteins to include experimentally un-characterized proteins that have either been indirectly associated with the known protein or share common domains or might localize to the same cellular compartment (https://genemania.org/) accessed on 19 October 2024 [14]. All these tools provided a powerful overview of functional connections between DEGs.

Using the Cytoscape software (version 3.10.2 accessed on 19 October 2024) platform for biological network analysis, generated networks were visualized and analyzed [15]. Fueled by the ability to visualize interactions using the software Cytoscape, critical hubs and functional clusters of network interactions were mapped. The Molecular Complex Detection (MCODE) algorithm was used to detect densely connected modules with regions of high connectivity that are often indicative of important protein complexes or co-regulated pathways. Based on these criteria, hubs and modules with high connectivity and biological relevance were prioritized further for in-depth analysis, given the potential involvement of these properties in TNBC pathogenesis and therapy resistance. The integrative PPI analysis generated a comprehensive molecular interaction map that provided potentially significant nodes and pathways that may act as biomarkers or points of therapeutic intervention and should be further validated experimentally.

### 2.3. Pathway Enrichment Analysis

The biological functions and clinical relevance of the differentially expressed genes (DEGs) were studied through pathway enrichment analysis and disease–drug association using Enrichr accessed between 17–22 October 2024, a comprehensive enrichment analysis web-based platform (https://maayanlab.cloud/Enrichr/) [16]. Pathway enrichment analysis was at the core of unraveling the functional implication of DEGs. Enrichr combines datasets from multiple databases to provide a granular view of enrichment across biological parameters.

### 2.4. Gene Ontology (GO)

GO enrichment analysis was performed in three key domains:GO Biological Processes (BP): This category exhibited cellular processes affected by DEGs, including apoptosis, immune responses, and mechanisms associated with tumor progression as the arms of importance in TNBC.GO Molecular Functions (MF): Gene product functions, including binding affinities, catalytic activities, and roles in signaling cascades, were identified through the analysis.GO Cellular Components (CC): Functional subclass distinguishes gene products, focusing on nuclear factors and membrane-bound proteins that appear central to TNBC pathology.

### 2.5. KEGG (Kyoto Encyclopedia of Genes and Genomes)

KEGG pathway analysis mentioned significant signaling pathways, such as those related to cancer development, immune system regulation, and ferroptosis, a form of programmed cell death that has drawn growing interest in TNBC (https://maayanlab.cloud/Enrichr/) [16]. These findings illuminate molecular networks mediating TNBC pathobiology.

### 2.6. Reactome

The KEGG pathway analysis provided a broad overview of enriched signaling networks, whereas Reactome delivered a higher-resolution mapping of the molecular interactions and biological processes implicated in TNBC. This combined approach identified statistically significant pathways (adjusted *p* < 0.05) associated with ferroptosis regulation, TNBC initiation, and anoikis resistance. Disease–gene association profiling was conducted using the Enrichr GWAS Catalog 2023 and DisGeNET databases to map ferroptosis-related DEGs to documented disease phenotypes. This yielded statistically significant associations with TNBC and other malignancies, indicating potential relevance of these genes to cancer biology. Drug–gene interaction mapping, performed via the DSigDB module in Enrichr, identified compounds with known modulatory effects on ferroptosis-related targets. Several of these compounds are already in clinical use or under development, providing a set of candidates for potential therapeutic repurposing. By integrating pathway enrichment with disease–gene and drug–gene interaction analyses, this workflow delineated a reproducible molecular profile for TNBC that may inform subsequent validation studies and the prioritization of therapeutic candidates.

### 2.7. Survival Analysis

The prognostic significance of ferroptosis-associated genes was assessed using two independent web-based platforms: BC-GenExMiner v4.10 (https://bcgenex.ico.unicancer.fr/BC-GEM/GEM-Accueil.php?js=1) and Kaplan–Meier Plotter (https://kmplot.com/analysis/) accessed on 25 October 2024 [17,18]. BC-GenExMiner integrates annotated transcriptomic datasets from GEO and ArrayExpress with corresponding clinical metadata for breast cancer subtypes, whereas Kaplan–Meier Plotter compiles harmonized expression and survival data from TCGA, GEO, and EGA repositories.

For each gene of interest, TNBC patient cohorts were dichotomized into high- and low-expression groups based on the median expression value. The survival endpoints evaluated included distant metastasis-free survival (DMFS), disease-free survival (DFS), and overall survival (OS). Kaplan–Meier survival curves were generated for each endpoint, and hazard ratios (HRs) with 95% confidence intervals (CIs) were calculated using the Cox proportional hazards regression model. Statistical significance was determined using the log-rank test, with *p*-values < 0.05 considered significant. All analyses used the platforms’ default settings for data normalization, patient stratification, and censoring.

## 3. Results

### 3.1. Differentially Expressed Genes (DEGs) and Common Genes Analysis

Differential expression analysis using the GEO2R platform identified a distinct set of genes significantly dysregulated in TNBC relative to normal breast tissue controls, several of which have established associations with ferroptosis. Based on curated functional annotations and expression behavior under ferroptosis-inducing conditions, these genes were classified into three categories: drivers, suppressors, and dual regulators.

The upregulated group comprised MAPK1, TLR4, ATM, ULK2, IFNG, and ATF3, all implicated in pathways related to oxidative stress signaling, DNA damage response, and immune regulation. The downregulated group included NFS1, GCLC, TP63, CD44, and SRC, genes involved in maintaining redox balance, membrane integrity, and iron–sulfur cluster biosynthesis. HMOX1 was identified as a dual regulator, encoding heme oxygenase-1, an enzyme responsible for heme degradation and modulation of intracellular iron availability.

This classification establishes a TNBC-specific ferroptosis-related gene expression profile, distinguishing consistently upregulated drivers, downregulated suppressors, and a context-dependent regulator.

### 3.2. Protein–Protein Interaction (PPI) Network Analysis

To gain deeper insight into the regulatory interactions among ferroptosis-associated genes in triple-negative breast cancer (TNBC), a protein–protein interaction (PPI) network was constructed using the STRING and GeneMANIA databases (Figure 1). This systems-level approach enabled the visualization of functional connections among ferroptosis drivers, suppressors, and dual regulators, offering mechanistic context for their potential cooperative roles within the tumor microenvironment.

### 3.3. STRING Database Analysis

Within the STRING network, MAPK1 displayed high connectivity, indicating its role as a central node integrating oxidative stress signals, cell proliferation, and survival pathways. The broad interaction profile of MAPK1 suggests it may act as a regulatory checkpoint, influencing ferroptotic sensitivity in response to redox imbalances and extracellular cues.

HMOX1 was detected as differentially expressed in TNBC but displayed bidirectional regulation patterns across samples, preventing its exclusive classification as either upregulated or downregulated. Cross-referencing the DEG list with the curated ferroptosis gene set from FerrDb v2 verified its established association with ferroptotic pathways. Within the STRING-derived PPI network, HMOX1 demonstrated high connectivity, including direct interactions with FTH1 and LAMP2, and achieved a centrality score above the predefined hub-selection threshold. Based on these objective analytical criteria, HMOX1 was retained in the final ferroptosis-related gene network as a dual-functional biomarker.

Another key node, TP63, showed multiple connections to stress-responsive and redox-regulating genes. As a member of the p53 family, TP63 is implicated in regulating lipid metabolism and antioxidant defense, suggesting a potential contribution to ferroptosis initiation under oxidative stress conditions.

Collectively, the PPI network highlights a coordinated regulatory architecture in TNBC, where MAPK1, HMOX1, IFNG, and TP63 converge at the interface of ferroptosis, immune modulation, and redox control. These hub genes may serve as strategic points for further experimental validation and functional exploration in ferroptosis-based therapeutic targeting strategies.

### 3.4. GeneMANIA Analysis

To complement the STRING-based network analysis, GeneMANIA was employed to investigate additional functional associations among ferroptosis-related genes in triple-negative breast cancer (TNBC). This expanded network, derived from co-expression patterns, co-localization data, shared protein domains, and curated pathway relationships, revealed further molecular interactions not captured in the primary STRING model (Figure 2)

Key regulators of iron metabolism, including HMOX1, FTH1, and LAMP2, were also integrated within the GeneMANIA network. The central placement of HMOX1 aligns with its known dual role in maintaining redox balance and modulating labile iron availability, both of which are essential in the ferroptotic process.

When combined with the STRING results, the GeneMANIA analysis emphasized MAPK1, TP63, and HMOX1 as central hub genes, further implicating them in ferroptosis regulation. The presence of additional transcriptional mediators such as FOS, JUN, and ATF3 supports the idea that ferroptosis in TNBC may be influenced by coordinated stress-responsive transcriptional networks.

In summary, the GeneMANIA analysis highlights a multilayered regulatory architecture in TNBC, involving both ferroptosis core effectors and upstream transcriptional stress regulators. These computational predictions offer mechanistic insights but will require functional validation to confirm their roles in ferroptosis susceptibility and clinical relevance.

### 3.5. Pathway Enrichment Analysis Findings

To evaluate the functional relevance of the differentially expressed genes (DEGs) identified in this study, enrichment analysis was conducted using Enrichr, focusing on Gene Ontology (GO) classifications. This included analyses of Biological Processes, Cellular Components, and Molecular Functions (Figure 3), with an emphasis on pathways related to ferroptosis regulation.

### 3.6. Gene Ontology (GO) Enrichment Analysis

In the Biological Process category, enriched terms such as regulation of autophagy, cellular response to oxidative stress, and mitochondrial disassembly suggest involvement of these genes in ferroptosis-associated stress responses and metabolic reprogramming in TNBC.

Enrichment in the Cellular Component category included terms like autophagosome, endoplasmic reticulum membrane, and mitochondrial outer membrane, indicating subcellular sites where ferroptosis-associated molecular events may occur.

For Molecular Functions, significant terms such as serine/threonine kinase activity, MAP kinase activity, and ubiquitin protein ligase binding highlight possible roles in post-translational modification and signal transduction pathways relevant to ferroptotic regulation.

Overall, these enrichment profiles support the involvement of the identified genes in ferroptosis-related processes in TNBC and offer mechanistic insight into their cellular roles.

### 3.7. Disease/Drug Association Analysis

To assess the clinical significance of the identified ferroptosis-related differentially expressed genes (DEGs), disease and drug association analyses were performed using Enrichr’s integrated databases: GWAS Catalog 2023, DisGeNET, and DSigDB. These platforms facilitated the identification of disease phenotypes and pharmacological agents potentially linked to ferroptosis regulation in TNBC (Figure 4).

The GWAS Catalog analysis demonstrated enrichment in hematologic parameters, including mean corpuscular hemoglobin concentration and transferrin saturation, indicating that the DEG set is functionally connected to iron metabolism, an essential determinant of ferroptotic susceptibility.

In the DisGeNET dataset, enriched associations included prostate cancer, chronic myeloid leukemia, and colon cancer, implying that ferroptosis-associated genes may have broader oncogenic relevance beyond TNBC and could represent shared regulatory axes across multiple tumor types.

The DSigDB drug–gene interaction analysis identified compounds such as deferoxamine and sulfasalazine, both of which influence iron metabolism or redox balance. Deferoxamine has been extensively validated as a potent ferroptosis inhibitor, acting through chelation of labile iron pools and consequent suppression of iron-dependent lipid peroxidation. In contrast, sulfasalazine is recognized for its ability to induce ferroptosis by inhibiting the cystine/glutamate antiporter system Xc^−^, thereby depleting intracellular glutathione and enhancing lipid ROS accumulation. These mechanistic distinctions are consistent with recent evidence highlighting the divergent yet complementary roles of iron chelators and system Xc^−^ inhibitors in ferroptosis modulation [19,20].

### 3.8. Reactome and KEGG Pathway Enrichment Analysis

To contextualize the biological functions of the identified ferroptosis-related genes, pathway enrichment analysis was performed using the Reactome and KEGG databases (Figure 5).

The Reactome analysis demonstrated significant enrichment in pathways related to autophagy, macroautophagy, and immune system modulation. These findings indicate that ferroptosis regulation in TNBC may be closely linked to autophagic degradation processes, which facilitate the removal of damaged organelles and maintain cellular homeostasis under stress. The association with immune-related pathways further suggests that ferroptotic signaling could intersect with tumor–immune interactions, potentially influencing immune evasion or activation dynamics.

In the KEGG analysis, the ferroptosis pathway was prominently enriched, validating the central involvement of the selected genes in this form of regulated cell death. Additionally, enrichment of the FoxO signaling pathway and the AGE–RAGE signaling pathway, both associated with oxidative stress, metabolic regulation, and inflammation, supports the view that ferroptosis is integrated within broader stress response and metabolic networks in TNBC pathophysiology.

### 3.9. Wiki Pathways Analysis

To further investigate the biological roles of ferroptosis-associated genes in TNBC, pathway enrichment analysis was performed using WikiPathways. This analysis provided complementary insights into how ferroptosis may interface with broader cellular processes, including oncogenic signaling, immune modulation, and stress response pathways (Figure 5).

Enriched pathways such as “Ferroptosis in Pancreatic Ductal Adenocarcinoma” and the “RAC1–PAK1–p38–MMP2 signaling axis” suggest that similar ferroptosis-associated mechanisms may be conserved across multiple tumor types. In addition, the enrichment of the “Host–Pathogen Interaction of Human CoVs via Autophagy” pathway indicates a potential link between ferroptosis and autophagy-driven responses to external stressors, including infectious agents.

These findings are consistent with prior enrichment results from the KEGG and Reactome databases, collectively reinforcing the association of ferroptosis with autophagy, oxidative stress regulation, and immune-related signaling pathways. The integration of these data highlights the involvement of ferroptosis in stress-adaptive cellular networks and underscores its potential relevance to TNBC biology.

### 3.10. Survival Analysis Findings

Survival analysis was performed to assess the prognostic significance of ferroptosis-related genes in triple-negative breast cancer (TNBC) using two independent platforms: BC-GenExMiner and the Kaplan–Meier Plotter. Analyses focused on three clinical endpoints: distant metastasis-free survival (DMFS), disease-free survival (DFS), and overall survival (OS).

### 3.11. BC Genex Analysis

The analysis utilized BC GenExMiner, an integrated platform incorporating transcriptomic and clinical data across breast cancer subtypes. Survival stratification based on gene expression levels allowed for the identification of candidate biomarkers associated with favorable outcomes in TNBC-specific cohorts.

In BC-GenExMiner, higher IFNG expression was significantly associated with prolonged DMFS, DFS, and OS in TNBC-specific cohorts. Validation using the Kaplan–Meier Plotter confirmed these associations. For DMFS, patients with high IFNG expression exhibited a hazard ratio (HR) of 0.51 (95% confidence interval [CI]: 0.37–0.69; *p* < 0.0001). For DFS, the HR was 0.55 (95% CI: 0.44–0.69; *p* < 0.0001), and for OS, the HR was 0.61 (95% CI: 0.47–0.78; *p* = 0.0001).

Kaplan–Meier survival curves were generated to assess the relationship between gene expression levels and clinical outcomes, specifically distant metastasis-free survival (DMFS), disease-free survival (DFS), and overall survival (OS), in patients with triple-negative breast cancer (TNBC) (Figure 6 and Figure 7).

Collectively, the survival analyses validate IFNG as a strong prognostic indicator in TNBC and support the broader hypothesis that ferroptosis-related genes may hold clinical relevance in guiding therapeutic decision making and stratified patient management.

## 4. Discussion

This study presents a comprehensive bioinformatics-based investigation into the ferroptosis landscape of triple-negative breast cancer (TNBC), with a central focus on the role of interferon-gamma (IFNG) as a potential prognostic marker and modulator of ferroptotic signaling. Our integrative approach, which included differential expression analysis, protein–protein interaction (PPI) mapping, pathway enrichment analysis, and survival correlation, reveals a complex regulatory network implicating ferroptosis in TNBC pathogenesis and progression. The findings contribute to a growing body of evidence positioning ferroptosis as a promising therapeutic avenue, particularly for tumors that exhibit resistance to conventional treatments.

Ferroptosis is a regulated form of non-apoptotic cell death characterized by the accumulation of iron-dependent lipid peroxides and oxidative damage [21]. Unlike classical forms of cell death such as apoptosis or necroptosis, ferroptosis is driven by metabolic dysregulation, impaired antioxidant defenses, and iron overload [22]. Central to this process are molecular regulators such as GPX4, which detoxifies lipid hydroperoxides; SLC7A11, a component of the cystine/glutamate antiporter that maintains glutathione levels; and iron-handling proteins like FTH1 and TFRC [23]. In the context of TNBC, which is both metabolically active and genetically unstable, the induction of ferroptosis represents a compelling therapeutic strategy to exploit cancer-specific vulnerabilities.

Importantly, several key ferroptosis drivers and suppressors, including MAPK1, TLR4, ATM, ULK2, NFS1, and HMOX1, have been identified, offering potential therapeutic targets to modulate cancer cell susceptibility to ferroptosis. One direct modulator of cell survival signaling pathways is MAPK1, which served as a potent target for inducing ferroptosis and overcoming apoptosis resistance in cancer cells [24]. Suppressing MAPK1, a mutated variant of the NF2-Ras signaling pathway, may increase the sensitivity of cancer cells to ferroptosis, which can eventually lead to targeted cell death in therapy-resistant cancers.

Our study identified a suite of ferroptosis-related genes with potential regulatory roles in TNBC. Among the most prominent were MAPK1, TLR4, ULK2, ATM, NFS1, and HMOX1 genes associated with stress response, immune regulation, and autophagy. For instance, MAPK1 participates in both cell proliferation and ferroptotic susceptibility. Its inhibition has been shown to enhance ferroptosis by disrupting cell survival signaling, a mechanism particularly relevant in cancers harboring Ras or NF2 mutations [21]. Similarly, TLR4, a toll-like receptor involved in innate immunity, may intersect with ferroptotic pathways via modulation of inflammatory ROS production [24].

Notably, HMOX1 (heme oxygenase 1) emerged as a key node with a dual role. In some contexts, HMOX1 promotes ferroptosis by releasing free iron through heme catabolism, thereby accelerating lipid peroxidation [25]. In others, it acts cytoprotectively by mitigating ROS levels and preserving redox balance. This context-specific behavior emphasizes the need for careful molecular stratification before considering HMOX1 as a therapeutic target. These findings collectively highlight the intricate interplay between ferroptosis and stress-adaptive responses in the TNBC tumor microenvironment [26].

The elevated expression of MAPK1, TLR4, ATM, ULK2, IFNG, and ATF3 identified in this study may reflect an increased intrinsic susceptibility of TNBC cells to ferroptotic induction under oxidative or therapeutic stress. Conversely, the downregulation of NFS1, GCLC, TP63, CD44, and SRC suggests a potential attenuation of antioxidant defense systems, thereby predisposing tumor cells to lipid peroxidation-mediated cell death. The dual functional nature of HMOX1 capable of facilitating ferroptosis through enhanced labile iron release or exerting cytoprotective effects by limiting oxidative damage further underscores the intricate, context-dependent regulation of ferroptosis in TNBC [27].

STRING network analysis further delineated a set of highly connected hub genes, including MAPK1, HMOX1, IFNG, and TP63, which form a coordinated regulatory architecture at the interface of oxidative stress, immune modulation, and redox balance. The extensive connectivity of MAPK1 positions it as a central integrator of proliferative and oxidative signaling pathways, potentially modulating ferroptotic sensitivity. TP63, through its associations with lipid metabolism and antioxidant defense genes, may contribute to initiating ferroptosis under oxidative conditions. HMOX1’s central placement and interactions with iron-regulatory genes such as FTH1 and LAMP2 reinforce its dual, context-dependent role in ferroptotic regulation.

Our PPI network analysis supports this connection, showing that IFNG interacts with multiple ferroptosis-relevant genes, many of which are also involved in immune modulation, oxidative signaling, and cell death regulation. These interactions suggest that IFNG may act as both a biomarker and functional effector in the ferroptotic response. Therapeutically, this opens the door to combination strategies that exploit IFNG-induced sensitization to ferroptosis, particularly in tumors exhibiting high IFNG expression or immune infiltration [28].

Moreover, the clinical implications extend to the potential use of biomarkers of iron metabolism such as transferrin saturation and mean corpuscular hemoglobin concentration as indicators of ferroptosis susceptibility [23,29]. Such markers may inform patient stratification for ferroptosis-inducing therapies, facilitating more personalized approaches to TNBC management.

Survival analyses performed using BC GenExMiner and the Kaplan–Meier Plotter platform demonstrated a consistent and statistically significant association between high IFNG expression and improved distant metastasis-free survival, disease-free survival, and overall survival in TNBC patients. The reproducibility of these findings across independent analytical platforms strengthens the proposition of IFNG as a robust prognostic biomarker. This consistency suggests that ferroptosis-related gene expression patterns could be leveraged for patient risk stratification and may guide the rational development of targeted therapeutic strategies.

However, it is critical to acknowledge the limitations of this study. Our findings are derived entirely from in silico analyses, and while they are supported by robust statistical validation, they lack direct experimental confirmation [30]. Future studies should prioritize functional validation through gene knockdown or overexpression, ferroptosis assays in TNBC models, and mechanistic interrogation of IFNG signaling. Additionally, dual-role genes such as HMOX1 require empirical investigation within specific TNBC subtypes or immune contexts to clarify their utility as therapeutic targets [31].

Integrative clinical research that combines transcriptomic profiling, immunophenotyping, and ferroptosis pathway analysis will be essential to translate these findings into actionable therapies [32]. The synergistic application of ferroptosis inducers with immune checkpoint inhibitors or autophagy modulators holds particular promise for TNBC subtypes that exhibit poor responsiveness to conventional therapies [33,34].

In addition to its well-recognized role in orchestrating antitumor immune responses, prior experimental evidence indicates that IFNG can modulate ferroptotic susceptibility by downregulating SLC7A11 expression and enhancing lipid peroxidation in select cancer contexts [35,36]. Our bioinformatics analyses on TNBC align with these observations, suggesting a possible link between IFNG expression and ferroptosis-related molecular networks. Nevertheless, these associations should be regarded as hypothesis generating, as causality cannot be established from in silico data alone. The influence of IFNG on ferroptosis is likely to be highly context dependent, necessitating targeted experimental validation before clinical translation.

In conclusion, our study elucidates the ferroptosis regulatory landscape in TNBC and nominates IFNG as a prognostically and mechanistically significant gene that connects immune signaling to ferroptotic cell death. These insights contribute to the evolving paradigm of ferroptosis as a therapeutic target and underscore its potential role in future precision oncology strategies for TNBC [37,38].

## 5. Conclusions

This study offers an integrative, systems-level perspective on the regulatory landscape of ferroptosis in triple-negative breast cancer (TNBC), revealing ferroptosis as a promising therapeutic vulnerability in this aggressive and treatment-refractory malignancy. Through transcriptomic profiling and protein–protein interaction (PPI) network analysis, we identified a subset of ferroptosis-related genes such as MAPK1, HMOX1, and TP63 that exhibit potential involvement in tumor progression, oxidative stress regulation, and immune interaction. These molecular players form part of a highly interconnected network that supports the concept of ferroptosis as a dynamic and context-sensitive mode of regulated cell death.

Among these regulators, HMOX1 emerged as a particularly nuanced candidate due to its dual role in ferroptosis modulation. Depending on cellular context, HMOX1 may either exacerbate ferroptotic cell death by increasing labile iron or confer protection by mitigating reactive oxygen species. This complexity highlights the necessity of context-specific strategies when considering ferroptosis-inducing therapies, particularly in heterogeneous tumors such as TNBC.

A central finding of this investigation is the prognostic and mechanistic significance of interferon-gamma (IFNG). High IFNG expression was consistently associated with improved clinical outcomes including overall survival and metastasis-free survival across TNBC cohorts. In addition to its well-established role in orchestrating antitumor immunity, IFNG may act as a sensitizer of ferroptosis through the downregulation of SLC7A11 and disruption of glutathione homeostasis. This dual functionality positions IFNG as a potential biomarker for ferroptosis responsiveness and a candidate for combinatorial immuno-ferroptotic therapeutic strategies.

Furthermore, enrichment analysis underscored the involvement of ferroptosis-associated genes in immune pathways, autophagy, and cellular metabolism, suggesting that ferroptosis is intricately linked to broader tumor microenvironment dynamics. These findings support the rationale for integrating ferroptosis-based treatments with established immunotherapies, such as immune checkpoint blockade, particularly in immunologically active TNBC subtypes.

While the findings presented here are statistically robust, it is important to acknowledge that the study is based solely on computational analyses. As such, these results should be considered hypothesis generating rather than conclusive. To advance these observations toward clinical relevance, experimental validation will be essential. Future studies should include in vitro and in vivo functional assays, genetic perturbation of candidate regulators, and prospective correlation with patient response to ferroptosis-targeted interventions.

In conclusion, this study provides a foundational framework for understanding ferroptosis regulation in TNBC and introduces IFNG as a dual-role immunological and ferroptotic modulator with significant translational potential. By elucidating the complex interplay between ferroptosis, immune signaling, and tumor progression, these findings offer new avenues for biomarker discovery and therapeutic innovation in TNBC and potentially other difficult-to-treat cancers characterized by immune resistance and metabolic adaptability.

## Figures and Tables

**Figure 1 biomedicines-13-02037-f001:**
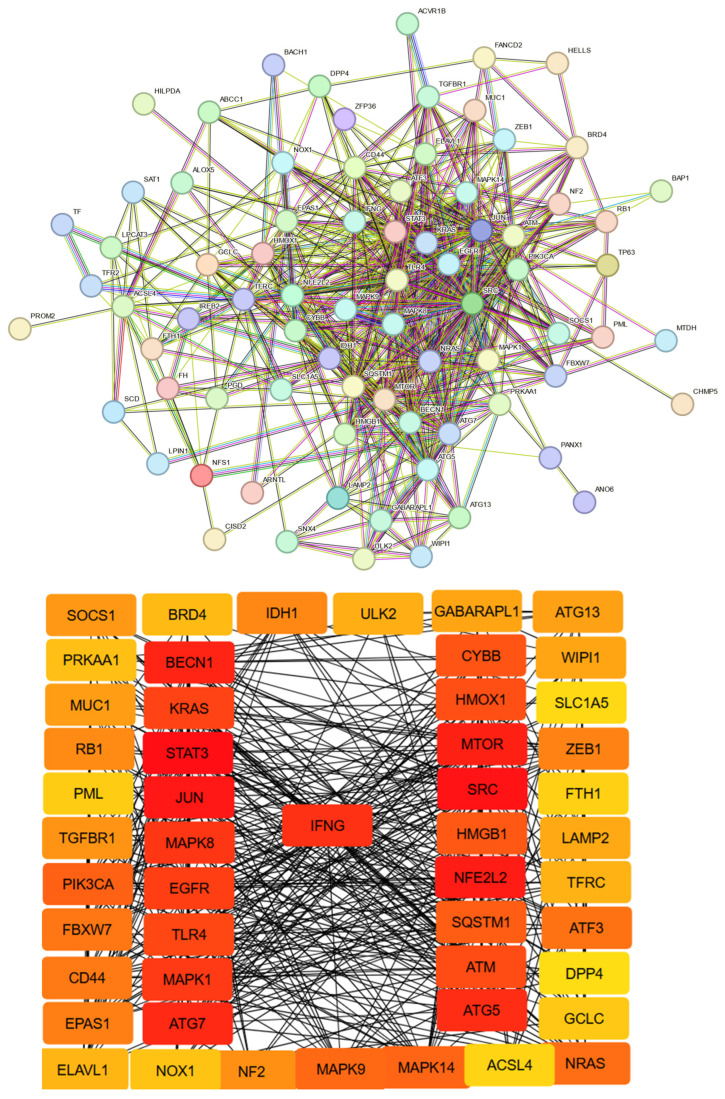
Protein–protein interaction (PPI) network for ferroptosis drivers, suppressors, and common genes was established using STRING. Key central nodes identified in the network include IFNG, MAPK1, HMOX1, and TP63.

**Figure 2 biomedicines-13-02037-f002:**
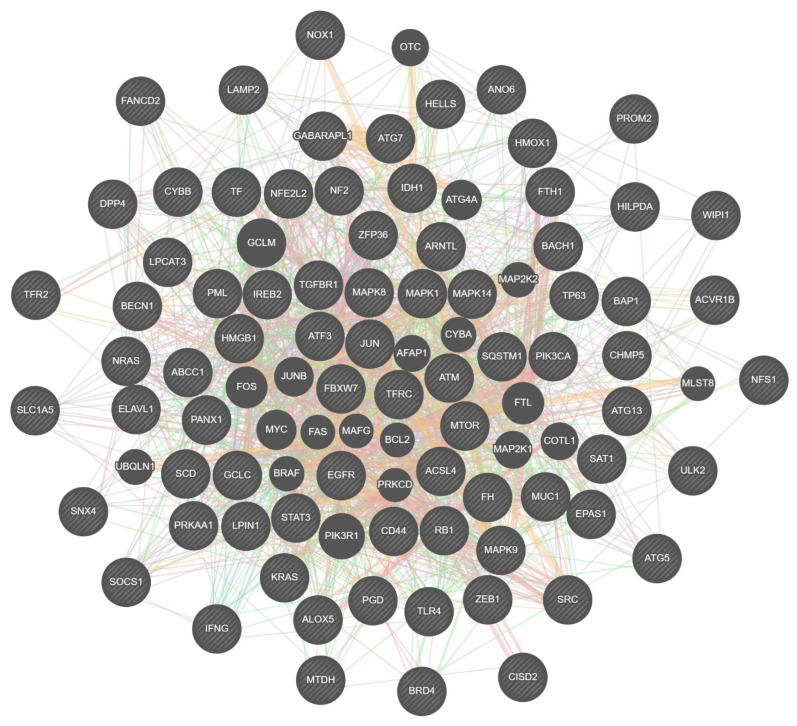
GeneMANIA network shows the relationships and functional associations between ferroptosis-related genes, highlighting the interlinked functions in stress response, iron metabolism and ferroptosis regulation.

**Figure 3 biomedicines-13-02037-f003:**
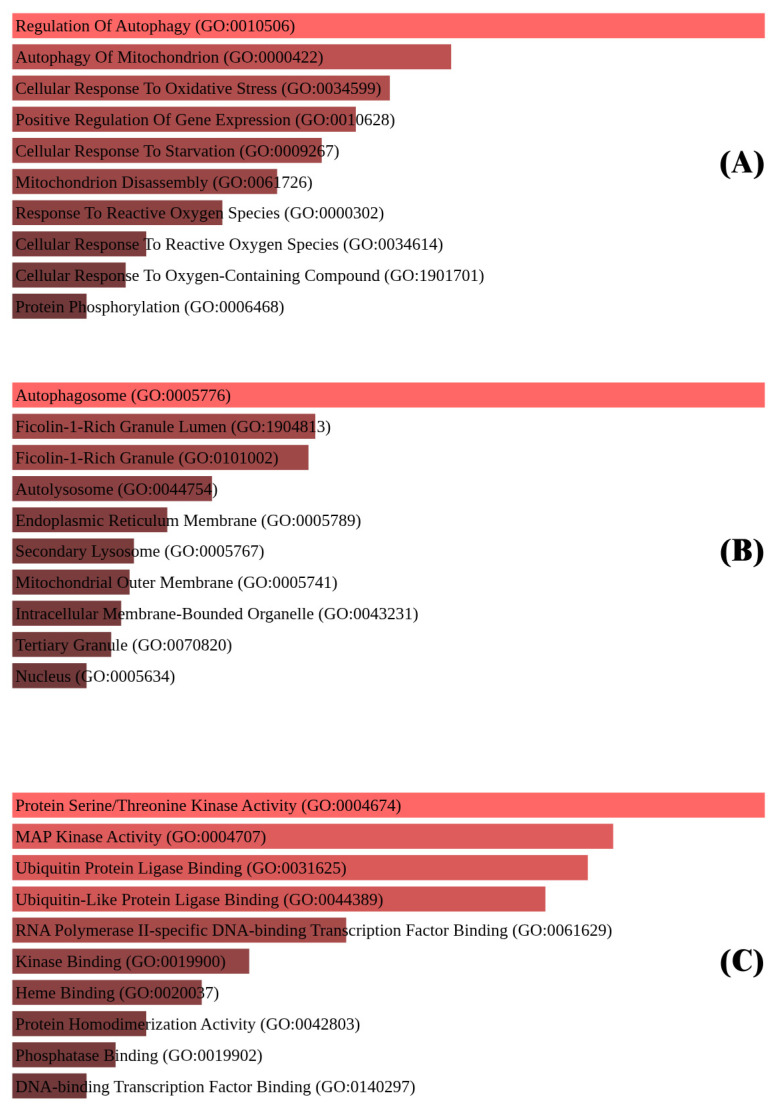
(**A**) Biological Processes: enrichment in oxidative stress responses, autophagy regulation, and cellular response to starvation. (**B**) Cellular Components: genes linked to autophagosomes, mitochondrial outer membranes, and endoplasmic reticulum. (**C**) Molecular Functions: key activities include serine/threonine kinase activity, MAP kinase signaling, and ubiquitin protein ligase binding.

**Figure 4 biomedicines-13-02037-f004:**
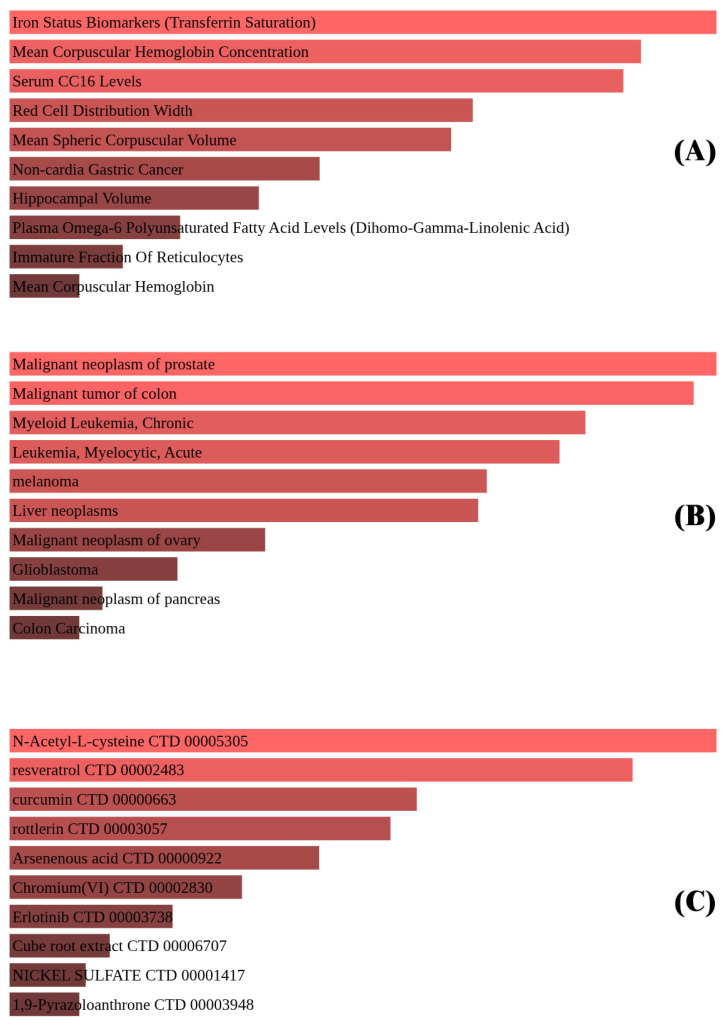
(**A**) Iron metabolism markers such as transferrin saturation and hemoglobin concentration. (**B**) Associations with cancers like prostate, colon, and glioblastoma. (**C**) Ferroptosis-modulating drugs, including N-acetyl-L-cysteine, resveratrol, and curcumin.

**Figure 5 biomedicines-13-02037-f005:**
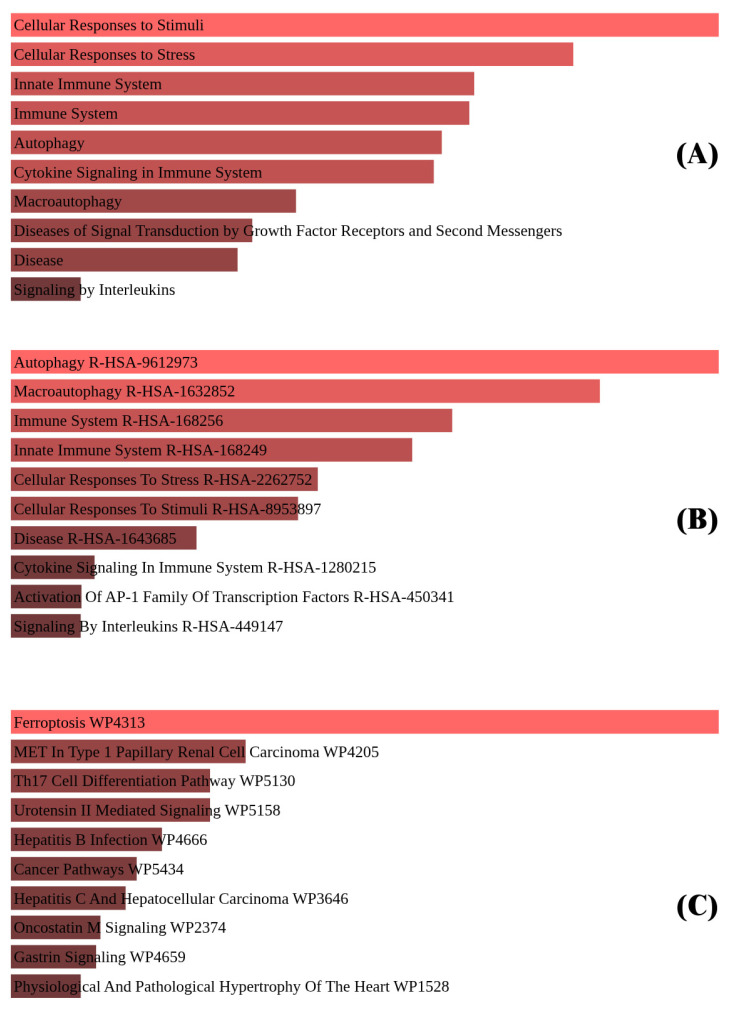
(**A**) Biological pathways include cellular responses to stimuli, immune system processes, and autophagy. (**B**) Disease pathways highlight ferroptosis, PD-L1 expression, and FOXO signaling. (**C**) Specific ferroptosis-related pathways such as RAC1 PAK1 P38 signaling and cancer pathways are identified as potential therapeutic targets.

**Figure 6 biomedicines-13-02037-f006:**
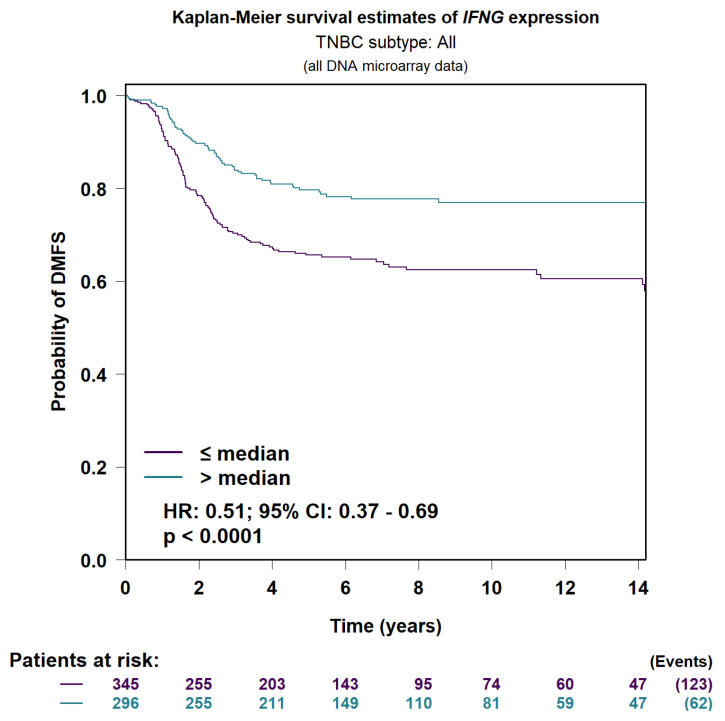
Kaplan–Meier survival curves using DMFS for TNBC patients scored by levels of IFNG expression, indicating improved survival achieved in the subgroup of patients displaying very high IFNG expression levels.

**Figure 7 biomedicines-13-02037-f007:**
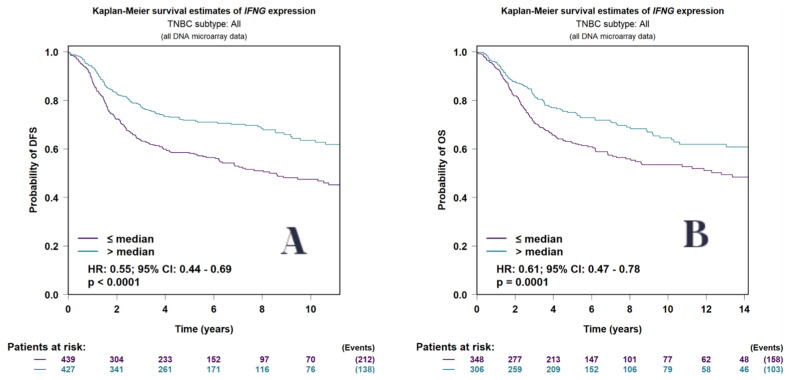
Kaplan–Meier survival estimates for TNBC patients based on IFNG expression levels. (**A**) Disease-free survival (DFS): patients with IFNG expression levels above the median showed significantly improved survival rates compared to those below the median (HR: 0.55; 95% CI: 0.44–0.69; *p* < 0.0001). (**B**) Overall survival (OS): high IFNG expression levels were associated with better overall survival (HR: 0.61; 95% CI: 0.47–0.78; *p* = 0.0001).

## Data Availability

The transcriptomic dataset analyzed in this study was obtained from the NCBI Gene Expression Omnibus (GEO) under accession number GSE103091 (https://www.ncbi.nlm.nih.gov/geo/query/acc.cgi?acc=GSE103091, accessed on 17 October 2024). Functional enrichment analyses, including Gene Ontology (GO), WikiPathways, Reactome, KEGG pathway enrichment, and disease/drug association analyses, were conducted using the Enrichr platform (https://maayanlab.cloud/Enrichr/, accessed between 17–22 October 2024). Protein–protein interaction (PPI) networks were constructed using STRING v11.5 (https://string-db.org/, accessed on 19 October 2024). Survival analyses were performed using Breast Cancer Gene-Expression Miner v4.11 (BC-GenExMiner) (https://bcgenex.ico.unicancer.fr/BC-GEM/GEM-Accueil.php?js=1, accessed on 25 October 2024) and the Kaplan–Meier Plotter (https://kmplot.com/analysis/, accessed on 25 October 2024). All datasets analyzed in this study are publicly available, and no new datasets were generated.

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
