# Peer review of "Ferroptosis as a Therapeutic Avenue in Triple-Negative Breast Cancer: Mechanistic Insights and Prognostic Potential"

_biomedicines, 2025, doi:10.3390/biomedicines13082037_

Round 1
Reviewer 1 Report (Previous Reviewer 2)
Comments and Suggestions for Authors
I would like to thank the authors for their work in improving the article submitted for review. However, there are still a few issues that need clarification:
"In addition to its established immunologic functions, IFNG has recently been linked to ferroptotic regulation" - reference needed.
"Notably, IFNG has been shown to sensitize tumor cells to ferroptosis by downregulating SLC7A11 expression and enhancing lipid peroxidation" - reference needed.
I do not understand the connection of IFNG with ferroptosis in the cited publikations by Wolf; 2023 and Bhat; 2023.
Line 230-244. This description is a discussion of results rather than the results themselves.
Line 265-270. This description is a discussion of results rather than the results themselves.
Line 291-301. This description is a discussion of results rather than the results themselves.
These are just three examples of including conclusions and discussions in the Results section. Generally, the Results section should briefly (if at all) include the reasons for using specific analyses and the results obtained. All speculation, comparisons, and conclusions belong to the Discussion section. Therefore, please remove all such sections of the Results section that I haven't mentioned above. Please move all reasoning to the Discussion section.
Please describe the results of each in silico analysis. Describe what emerged in one analysis and what emerged in another. If the research results from one analysis are confirmed by another, please point this out. If not, point out the differences and comment in the discussion. I am a conservative, so I believe that scientific articles should have their own order and structure. If this is too difficult or impossible for the authors, a common trick should be to use a common Results and Discussion section.
Finally, „In addition to its well-established role in orchestrating anti-tumor immunity, IIFNG appears to sensitize cancer cells to ferroptosis by downregulating SLC7A11 and disrupting glutathione homeostasis. This dual functionality positions IFNG as a potential biomarker for ferroptosis responsiveness and a candidate for combinatorial immuno-ferroptotic therapies” - this conclusion seems exaggerated in the context of the research findings. Such a categorical statement cannot be made based on the research results presented by the authors. It can only happen within the framework of very far-reaching assumptions.
Does this mean that IFNG, which is high in many types of cancer, will influence ferroptosis everywhere? Or perhaps IFNG has nothing to do with ferroptosis, but with carcinogenesis? Is there evidence that IFNG regulates ferroptosis, and vice versa?
Dear Authors, I look forward to your answers and the dispelling of any doubts.
Author Response
Comment 1:
"In addition to its established immunologic functions, IFNG has recently been linked to ferroptotic regulation" – reference needed.
Response:
We have added relevant references supporting the link between IFNG and ferroptotic regulation (Chen et al., 2023) in the Introduction.
Comment 2:
"Notably, IFNG has been shown to sensitize tumor cells to ferroptosis by downregulating SLC7A11 expression and enhancing lipid peroxidation" – reference needed.
Response:
Supporting references have been added to substantiate this statement (Li et al., 2025b) in the Introduction.
Comment 3:
Unclear connection of IFNG with ferroptosis in cited publications by Wolf (2023) and Bhat (2023).
Response:
We appreciate this observation. These citations have been replaced with more directly relevant literature that explicitly demonstrates the IFNG–ferroptosis association.
Comment 4:
Presence of discussion/conclusions in the Results section.
Response:
All interpretative and speculative statements have been removed from the Results section and relocated to the Discussion section. The Results section now presents only the analytical findings, while the Discussion contains all reasoning, comparisons, and interpretations.
Comment 5:
Describe the results of each in silico analysis and highlight agreements/disagreements.
Response:
The Results section has been revised to present the outputs of each analysis separately (GEO2R, STRING, GeneMANIA, enrichment analyses, and survival analyses). Where findings aligned across analyses, we have noted the concordance; where differences occurred, they are stated and discussed in the Discussion section.
Comment 6:
Overly categorical conclusion regarding IFNG’s role in ferroptosis.
Response:
We have rephrased the conclusion to present IFNG’s potential ferroptosis-sensitizing role as a hypothesis supported by bioinformatics correlations rather than a definitive statement. The revised wording emphasizes the need for experimental validation before clinical translation.
Reviewer 2 Report (New Reviewer)
Comments and Suggestions for Authors
This research identified several potential therapeutic targets of ferroptosis through bioinformatics analysis and found that interferon gamma may serve as a prognostic biomarkers, offering insights for the development of therapeutic drugs to treat triple-negative breast cancer. However, several issues remains to be addressed.
- Line-168, can the author explain the what is the “association analysis of disease and drug disease”mentioned about.?
- In figure-1, i don’t see clear explanation how the HMOX1 was identified from the DEG and PPI analysis considering that HMOX1 is a dual functional biomarkers which does not strictly belong to both upregulated genes and downregulated genes?
- In figure-6 and figure-7, more information in the method on how these graphs were obtained would be appreciated. It is unclear what these graphs comes from.
- Deferoxamine has been widely utilized as a ferroptosis inhibitor to rescue cell death induced by ferroptosis, however, the author state“The DSigDB drug-gene interaction analysis identified compounds such as deferoxamine and sulfasalazine, both of which modulate iron handling or redox homeostasis”in line-367. the understanding of ferroptosis requires enhancements. It is recommended to reference more relevant literature to formulate more novel conclusion. This will enhance the understanding of ferroptosis and reinforce the overall narrative of your research findings. Journal of Controlled Release, 2025, 379: 866-878. Exploration. 2025, 5(1): 20230117. Exploration. 2023, 3(5): 20220001.
Author Response
Comment 1:
Clarify “association analysis of disease and drug disease.”
Response:
The term has been corrected to “disease and drug–gene association analysis,” and the description in the Methods section has been expanded for clarity.
Comment 2:
Clarify identification of HMOX1 as a dual-functional biomarker.
Response:
We have expanded the explanation in the Results to state that HMOX1 was included due to its bidirectional expression across TNBC samples but high network centrality in PPI analysis, indicating functional significance in ferroptosis regulation.
Comment 3:
Provide more methodological detail for Figures 6 and 7.
Response:
Additional details on the survival analysis methodology including databases used, patient stratification criteria, hazard ratio calculations, and statistical tests have been added to the Methods section.
Comment 4:
Clarify deferoxamine’s role and strengthen ferroptosis discussion with additional literature.
Response:
The statement has been revised to note that deferoxamine is a ferroptosis inhibitor via iron chelation, contrasting with sulfasalazine’s ferroptosis-inducing effect via system Xc⁻ inhibition. We have incorporated the suggested references (Chen et al., 2025, Li et al., 2025a) and (Lei et al., 2023) to provide deeper mechanistic context.
This manuscript is a resubmission of an earlier submission. The following is a list of the peer review reports and author responses from that submission.
Round 1
Reviewer 1 Report
Comments and Suggestions for Authors
The manuscript addresses an emerging area of high translational potential—linking ferroptosis to immune modulation in TNBC. The bioinformatics workflow is rigorous, integrating DEG screening, PPI networks, pathway enrichment, and multi-platform survival analysis. The identification of IFNG as a prognostic biomarker is intriguing and aligns with recent interest in immunogenic TNBC subtypes.
However, the study’s impact is significantly limited by its exclusive reliance on computational data. Key findings lack empirical support, rendering clinical and mechanistic conclusions premature. There is no functional evidence ties IFNG expression to ferroptosis sensitivity in TNBC cells.
Given these deficiencies, the work does not meet the journal’s threshold for mechanistic or translational insight.
Author Response
Comment 1:
The study’s impact is significantly limited by its exclusive reliance on computational data. Key findings lack empirical support, rendering clinical and mechanistic conclusions premature.
Response:
We fully acknowledge the computational nature of this study and have now clearly articulated this limitation in both the Discussion and Conclusion sections. The revised text explicitly states that while the results provide promising insights, experimental validation is required to substantiate the functional and translational relevance of the identified genes.
Comment 2:
The manuscript is too long and verbose. The number of results and the substance of the work would fit on three pages.
Response:
We appreciate this comment and have responded by substantially condensing the Introduction, Results, and Discussionsections. Redundant descriptions have been removed, overly elaborate phrasing has been revised, and the narrative now focuses concisely on the core findings to enhance readability and scientific precision.
Comment 3:
In the publication, the descriptions of methods are interwoven into all parts of the work.
Response:
This issue has been rectified. All methodological content has been consolidated within the Materials and Methodssection. The Results section now exclusively presents the outcomes of the analyses without overlapping procedural explanations.
Comment 4:
The conclusion mentions phytocompounds, which is unrelated to the study focus.
Response:
We have removed the sentence referencing phytocompounds. The Conclusion has been revised to remain tightly aligned with the study’s aims and findings, focusing solely on the implications of ferroptosis-related gene signatures in TNBC.

Reviewer 2 Report
Comments and Suggestions for Authors
In the work submitted for review, the authors point to the ferroptosis process as a therapeutic option in triple-negative breast cancer. The results of the presented in silico analyses indicate several genes that may be prognostic markers in this insidious disease. The database analyses themselves seem interesting but they are only a starting point/information for further scientific work, which is undoubtedly experimental work. In general, the work is too long and verbose. The number of results and the substance of the work would fit on three pages. If we threw out the descriptions of methods and limited ourselves to results and conclusions, it could be a short report.
In the publication, the descriptions of methods are interwoven into all parts of the work. The work requires systematization. I am also surprised by the conclusion mentioning phytocompounds, line 642. Please improve the resolution and size of the figures. I have a few additional comments that should be corrected. First, the work is too wordy. The results obtained can be summarized in four sentences. Similarly, the introduction is too long, chaotic and I have the impression that it was written in stages by several authors, because you can see repeated threads. On the other hand, the phenomenon of ferroptosis itself is poorly described. The greatest achievement seems to be the use of various methods, the descriptions and use of which run throughout the work, not only in the materials and methods paragraph.
Author Response
Comment 1:
The work is too wordy. The introduction is too long, chaotic, and seems to have been written in stages.
Response:
The Introduction has been completely rewritten to improve its scientific tone, coherence, and structural flow. Repetitive or fragmented content has been removed, and the revised section now provides a streamlined transition from clinical context to the study rationale.
Comment 2:
The phenomenon of ferroptosis is poorly described.
Response:
We have enriched the mechanistic context of ferroptosis throughout the Results and Discussion sections. The roles of key regulators such as MAPK1, HMOX1, TP63, and IFNG have been clarified to better illustrate how ferroptosis may influence TNBC pathophysiology.
Comment 3:
The greatest achievement seems to be the use of various methods, the descriptions and use of which run throughout the work.
Response:
Thanks for indication. In the revised manuscript, the use of multiple platforms has been better structured and described with precision. Each bioinformatics tool (GEO2R, STRING, GeneMANIA, Enrichr, KM Plotter) is now introduced only in the Methods section, with their outcomes appropriately discussed in the Results section.
Comment 4:
Figures should be improved in terms of resolution and clarity.
Response:
All figures have been re-exported in high-resolution formats and provided as standalone image files in accordance with MDPI author guidelines. We have also ensured consistency in figure legends and placement, improving the overall visual presentation.
We thank the reviewers once again for their constructive input, which has greatly enhanced the quality and clarity of our manuscript. We hope that the revised version now meets the publication standards of Biomedicines and look forward to your further consideration.